# Autophagy as a Therapeutic Target of Natural Products Enhancing Embryo Implantation

**DOI:** 10.3390/ph15010053

**Published:** 2021-12-31

**Authors:** Hyerin Park, Minkyoung Cho, Yoonju Do, Jang-Kyung Park, Sung-Jin Bae, Jongkil Joo, Ki-Tae Ha

**Affiliations:** 1Department of Korean Medical Science, School of Korean Medicine, Yangsan 50612, Korea; rin8998@pusan.ac.kr (H.P.); uk0243@pusan.ac.kr (Y.D.); 2Korean Medical Research Center for Healthy Aging, Pusan National University, Yangsan 50612, Korea; letloves1@pusan.ac.kr (M.C.); Dr.NowOrNever@outlook.com (S.-J.B.); 3Department of Korean Medicine Obstetrics and Gynecology, Pusan National University Korean Medicine Hospital, Yangsan 50612, Korea; vivat314@pusan.ac.kr; 4Department of Obstetrics and Gynecology, Pusan National University Hospital, Busan 49241, Korea

**Keywords:** autophagy, embryo implantation, female infertility, natural products

## Abstract

Infertility is an emerging health issue worldwide, and female infertility is intimately associated with embryo implantation failure. Embryo implantation is an essential process during the initiation of prenatal development. Recent studies have strongly suggested that autophagy in the endometrium is the most important factor for successful embryo implantation. In addition, several studies have reported the effects of various natural products on infertility improvement via the regulation of embryo implantation, embryo quality, and endometrial receptivity. However, it is unclear whether natural products can improve embryo implantation ability by regulating endometrial autophagy. Therefore, we performed a literature review of studies on endometrial autophagy, embryo implantation, natural products, and female infertility. Based on the information from these studies, this review suggests a new treatment strategy for female infertility by proposing natural products that have been proven to be safe and effective as endometrial autophagy regulators; additionally, we provide a comprehensive understanding of the relationship between the regulation of endometrial autophagy by natural products and female infertility, with an emphasis on embryo implantation.

## 1. Introduction

Autophagy is a major pathway for lysosome-mediated degradation and recycling of a wide variety of biological macromolecules, including proteins, carbohydrates, lipids, and nucleic acids [1]. In the late 1950s, electron microscopy studies have contributed to the discovery of autophagy. Christian De Duve recognized a lysosome-dependent cellular process for the degradation of intracellular materials and termed it “autophagy” in 1963 [2]. In the early 1990s, Yoshinori Ohsumi created a new paradigm for understanding autophagy by identifying the essential genes for autophagy in baker’s yeast [3,4]. Since this breakthrough, molecular studies on autophagy have been conducted in mammalian cells as well as in yeast, and have fueled major advances in biomedical research [5].

Autophagy is well-established as an important mechanism for maintaining cellular homeostasis, including organelle integrity, stress response, metabolic regulation, protein quality control, and host defense, via the removal or recycling of intracellular molecules [1,5,6]. Emphasizing the importance of autophagy, various studies have suggested that defective autophagy contributes to and is a therapeutic target for multiple human diseases, such as asthma, systemic lupus erythematosus, Crohn’s disease, Parkinson’s disease, and several types of cancer [7,8,9,10,11]. Recent studies have revealed that autophagy also plays a fundamental role in male and female infertility by regulating the developmental process of reproductive organs and germ cells [12,13]. In particular, there is a specialized endometrial autophagy process to maintain processes that are vital to endometrial homeostasis, including menstruation, embryo implantation, and decidualization [14,15]. In addition, there is evidence that endometrial autophagy is essential for embryo implantation [16,17,18].

Embryo implantation is defined as a crucial process for attachment of the blastocyst, which is a properly developed embryo, to a receptive uterus and its implantation into the epithelium. Defective embryo implantation leads to an unsuitable environment for pregnancy, leading to a variety of additional problems, including infertility, subfertility, spontaneous miscarriage, abnormal intrauterine fetal growth, and pre-eclampsia [18,19]. Given the importance of embryo implantation, the regulation of endometrial receptivity is currently considered as one of the primary therapeutic strategies for treating female infertility, particularly repetitive implantation failure [19,20].

For successful embryo implantation, the endometrium must undergo various internal changes to increase receptivity for the embryo, without which embryo implantation will not occur or fail. Previous studies have shown that different synchronized molecular processes mediated by a variety of proteins, including cytokines, growth factors, adhesion molecules, and angiogenic factors, are required to regulate endometrial receptivity [21,22,23]. These molecular and cellular processes are commonly called endometrial autophagy, and defective endometrial autophagy results in endometrial hyperplasia, endometrial carcinoma, endometriosis, and infertility [15,24,25]. Therefore, understanding the role of autophagy in the endometrium holds promise for the development of novel therapeutic strategies for improving endometrial function.

Infertility is an important global public health problem, and the use of natural products to treat infertility has been considered a promising and safe alternative to conventional therapies [26,27,28]. Natural products are derived from diverse sources, including plants, bacteria, and fungi, and have been used as therapeutic candidates for various diseases, including Alzheimer’s disease, asthma, atherosclerosis, cancers, obesity, rheumatoid arthritis, and ulcerative colitis [29,30,31,32,33]. In addition, to better understand the efficacy of natural compounds against infertility and their safety, mechanistic studies are currently being conducted to discover natural substances that are effective in treating infertility.

To date, several studies have shown that natural products regulate endometrial autophagy and are effective as a treatment for infertility; however, there is no evidence of an association between their ability to control endometrial autophagy and improve infertility. Therefore, in this review, we discuss the correlation between the regulation of embryo implantation and endometrial autophagy after treatment with natural products. Additionally, this review suggests a new direction for research on the mechanism of infertility treatment with natural products.

## 2. Autophagy

Autophagy plays an important role in maintaining cellular homeostasis and energy levels. There are at least three forms of autophagy that depend on the cargo delivery system to the lysosome: macroautophagy, microautophagy, and chaperone-mediated autophagy (Figure 1). Macroautophagy, which is the primary autophagic pathway, refers to the formation of autophagosomes to collect cellular material, which subsequently fuses with lysosomes to break down the material. In contrast, in microautophagy, lysosomes directly engulf cytoplasmic cargo and degrade the material. Chaperone-mediated autophagy, which is unique to mammalian cells, involves a cargo recognition complex in the cytosol (heat shock protein 70 chaperone, Hsp70) and a cargo translation complex at the lysosome (lysosomal-associated membrane protein type 2A, Lamp2A). Hsp70 recognizes and attaches to a specific motif sequence in substrate proteins, and then the complex is delivered to Lamp2A in the lysosome membrane. The substrate protein is translocated to the lysosomal lumen for degradation by a lysosomal channel formed by Lamp2A [34].

Macroautophagy (hereafter referred to as autophagy) removes and recycles intracellular components, such as damaged organelles and unnecessary proteins. Autophagic activity is low under normal conditions; however, it increases with nutrient starvation, infection, and accumulation of unused components [35]. Defects in autophagy lead to the disruption of intracellular homeostasis and have been reported to cause various diseases, including metabolic and neurodegenerative diseases as well as various types of cancer [36]. Therefore, it is important to understand the molecular basis of autophagy. However, the physiological roles of autophagy are still not fully understood owing to a lack of methods for assessing autophagic flux. Therefore, the importance of quantitative assay systems for autophagic flux has been identified as a critical barrier to understanding autophagy as a therapeutic target for diverse diseases.

## 3. Regulation of Autophagosome Formation

Understanding the molecular basis of the formation and composition of cellular structures involved in autophagy is vital for improving our understanding of the process. As autophagy serves as a dynamic recycling machinery that maintains homeostasis for recycling cellular components and damaged organelles, the process is strictly regulated under physiological and pathological conditions [37]. The most important pathways for regulating autophagy are the mammalian target of rapamycin (mTOR) and AMP-activated protein kinase (AMPK) pathways [38,39]. AMPK, a key energy sensor and regulator of cellular metabolism, activates autophagy in response to ATP deficiency. Conversely, autophagy is inhibited by mTOR, a central cell-growth regulator that integrates growth factors and nutrition signals [39]. These two pathways counteract each other through phosphorylation of different serine residues of Unc-51 like kinase 1 (ULK1) or direct inhibition of mTOR1 by activated AMPK, thereby tightly regulating the initiation of autophagy [38,40]. Diverse stress conditions, such as nutrition starvation, growth factor deprivation, endoplasmic reticulum stress, viral infection, genotoxic stress, and oxidative stress, are known as physiological autophagy inducers [41,42,43]. Among the upstream regulators, liver kinase B1 (LKB1) is a master kinase of AMPK activation and serves as a metabolic checkpoint for cell growth in low nutrient conditions [44,45]. In addition, other stress conditions such as reactive oxygen species (ROS), which accumulate during glucose and amino acid deprivation, can also activate AMPK through activation of LKB1 or direct *S*-glutathionylation of cysteine residues on AMPK [42,46]. Other pathways, including p62/Keap1/Nrf2 and DNA damage response, also mediate the intercommunication between oxidative stress and autophagy [42].

Autophagosome formation can be divided into three stages: initiation, nucleation, and expansion (elongation). The process of autophagosome formation is shown in Figure 2. To initiate autophagy, the ULK complex (which contains the Ser/Thr kinases ULK1 and/or ULK2, autophagy-related protein 13 (Atg13), FAK family kinase-interacting protein of 200 kDa (FIP200), and Atg101) and the class III phosphoinositide 3-kinase (PI3K-III) complex, also known as the Beclin1 complex (which is composed of vacuolar protein sorting 34 (Vps34), p150, Beclin1, and Atg14), are required. In mammalian cells, the ULK complex is bound to mTOR complex 1 (mTORC1) and is inactive under fed conditions. Upon amino acid starvation, the ULK complex dissociates from mTORC1 and is activated, resulting in increased kinase activities of ULK1 and ULK2. Next, the ULK complex binds to the membrane, which is the site of autophagosome initiation, and recruits the complexes for autophagosome nucleation [47]. The ULK complex phosphorylates the PI3K-III complex, which is equally important for autophagosome initiation and activates Vps34 lipid kinase.

Following autophagosome initiation, Vps34 generates phosphatidylinositol 3,4,5-triphosphate (PI3P) on the membrane, which becomes a phagophore. These events drive the nucleation of the isolation membrane and recruit additional Atg proteins and autophagy-specific PI3P effectors, such as WD-repeat domain phosphoinositide-interacting 2 (Wipi2) and double FYVE-containing protein 1 (Dfcp1) [48]. During autophagosome nucleation, the interaction of PI3P with Wipi2 contributes to phagophore formation.

During expansion, the Atg12-Atg5-Atg16L1 complex (also known as the Atg16L1 complex) is recruited to the membrane, where it lipidates microtubule-associated protein 1 light chain 3 (MAP1-LC3, hereafter referred to as Lc3). Thus, Lc3 is associated with the autophagosomal membrane [49]. The association of cytosolic Lc3 proteins with the membrane occurs during the expansion of the isolation membrane. Before the closure of the isolation membrane, which becomes an autophagosome, the Atg proteins are dissociated from the membrane; however, lipidated Lc3 remains attached [50]. The Lc3 protein is thought to aid the expansion and closure of the isolation membrane [51,52] and its association with the autophagosomal membrane provides an important marker for identifying autophagosomes in cells.

There have been various suggestions regarding the origins of phagophore membranes and nucleation sites. These include de novo synthesis and pre-existing cellular membranes, such as the endoplasmic reticulum (ER), Golgi, mitochondria, endosomes, and the plasma membrane. However, recent data suggest that the ER is the most essential site for phagophore formation and elongation upon amino acid starvation, although the Golgi, mitochondria, plasma membrane, and endosomes also contribute to these events. The Golgi apparatus is essential for the trafficking of Atg9-containing vesicles and mitochondria supply lipid vesicles to the phagophore upon starvation. The plasma membrane is also a source of phagophore and autophagosome membranes under both basal and starvation conditions [47].

Lc3, a mammalian homolog of yeast Atg8, is widely used to measure autophagic activity. In humans, three paralogs of Lc3 have been reported: Lc3a, Lc3b, and Lc3c, which are encoded by the *MAP1LC3A*, *MAP1LC3B*, and *MAP1LC3C* genes, respectively. The cellular distribution, molecular function, and regulation of Lc3a and Lc3c have not yet been studied. Thus, Lc3b is commonly used in autophagy studies [53]. The Lc3 protein undergoes a series of post-translational modifications. The pro-form of Lc3b is cleaved at the carboxyl-terminal (C-terminal) by Atg4 and becomes cytosolic Lc3b-I, thereby exposing the C-terminal glycine residue. When autophagy is induced, Lc3b-I is subsequently transferred to the autophagosome by Atg3 and conjugated with phosphatidylethanolamine (PE) at the C-terminal glycine residue by the Atg16L1 complex, resulting in the formation of Lc3b-II [54]. The lipidated Lc3b-II is bound to both the outer and inner membranes of the autophagosome [55]. During the autophagy process, Lc3b-II bound to the autophagosomal inner membrane is degraded by lysosomal enzymes, whereas those located in the autophagosomal outer membrane are released into the cytosol and recycled [56]. Owing to this property, Lc3b is widely used as an autophagosome marker.

The fusion of autophagolysosomes with lysosomes is indispensable for the completion of the catabolic process of autophagy [57,58]. Transcription factor EB (TFEB), a member of the MiTF/TFE3 family, has been regarded as a master transcriptional regulator of lysosomal biogenesis [59,60]. The nuclear translocation and transcriptional activity of TFEB are controlled by the phosphorylation of specific serine residues. When serine residues including Ser122, Ser142, and Ser211 are phosphorylated, TFEB is inactive and localizes to the cytoplasm [61]. The Ser211 residue is crucial for binding to the 14-3-3 scaffold protein and subsequent cytoplasmic sequestration [62]. The phosphorylation of TFEB is mainly controlled by mTOR kinase and by phosphatase, protein phosphatase 3 catalytic subunit beta (PPP3CB) [59,63]. Other kinases, such as extracellular signal-regulated kinase (ERK), glycogen synthase kinase-3β (GSK3β), and Akt, are also involved in the phosphorylation and cytoplasmic retention of TFEB [60,64,65]. However, the regulatory role of mTOR in nuclear translocation of TFEB is controversial. The phosphorylation of Ser462, Ser463, Ser466, Ser467, and Ser469, which are located at the C-terminus of TFEB, drive its nuclear translocation. These residues can be phosphorylated by mTOR or protein kinase C (PKC) β [66,67]. TFEB translocated into the nucleus regulates the expression of target genes bearing the coordinated lysosomal expression and regulation (CLEAR) motif, thereby participating in the formation and lysosomal fusion of autophagosomes as well as lysosomal biogenesis [57,68]. Thus, TFEB activity has been regarded as a potential target for modulating autophagy and lysosomal function for treating several pathological conditions, including cancer and neurodegenerative diseases [59,69].

## 4. Regulation of Embryo Implantation

Pregnancy is a complex, but highly organized process that comprises multiple steps, including fertilization, implantation, decidualization, placentation, and the birth of offspring [70]. During the early stage of pregnancy, the endometrium undergoes major cellular changes, such as the receptiveness of the endometrial epithelium and decidualization of endometrial stromal cells (ESCs) [71]. The ability of the endometrium to allow embryo implantation is referred to as endometrial receptivity [18]. The features of endometrial receptivity include histological changes, such as angiogenesis, edema, and enhanced secretory activity of the endometrial glands [19,72]. Decidualization refers to significant changes occurring in uterine ESCs, including morphological and functional changes in ESCs, vascular changes to endometrial arteries, extracellular matrix remodeling, and the appearance of immune cells. Decidualization plays an important role in placental formation between the uterus and fetus by mediating the invasiveness of trophoblast cells [73].

These complex processes are regulated by diverse factors, including (1) the ovarian steroid hormones progesterone and estrogen, (2) the cytokines leukemia inhibitory factor (LIF) and interleukin 6 (IL6), and (3) growth factors such as transforming growth factor-β (TGF-β) and heparin binding-epidermal growth factor (HB-EGF). These factors regulate the expression of several integrin molecules. Integrin molecules play a crucial role in the attachment of blastocysts to the uterine epithelium [74]. During the implantation period, ovarian steroids facilitate appropriate morphology, function, and development of the endometrium [75]. The endometrium in the mid-to-late-secretory phase, where the concentrations of ovarian steroid hormones are highest and implantation occurs, shows high expression levels of cytokines such as LIF and IL6 [76,77]. Cytokines play an important role in the adhesion between the endometrium and embryo during implantation and promote placental development. In particular, diminished secretion of LIF is associated with recurrent implantation failure (RIF) [78]. TGF-β and HB-EGF are expressed in endometrial stromal and epithelial cells and have been reported to regulate endometrial cell proliferation and decidual transformation [74]. The major factors regulating endometrial receptivity and decidualization are summarized in Figure 3.

Although assisted reproductive technology (ART) has advanced, the implantation success rates of transferred embryos have not improved sufficiently [20,79]. A variety of studies, including those on growth factor treatment, immune therapy, platelet-rich plasma infusion, and intentional endometrial injury, have been conducted to improve the implantation rate [79,80,81,82,83]. However, there are very limited options for improving improper endometrial receptivity and decidualization [84]. Thus, more profound approaches are required to comprehend the molecular basis of embryo implantation and thereby identify novel therapeutics to improve the implantation rate.

## 5. Role of Autophagy in Embryo Implantation

Autophagy is a ubiquitous physiological process that plays diverse functions in different processes and diseases, both in stromal cells and epithelial cells in the endometrium [85,86,87]. Previously, Peters et al. [12] reviewed the role of autophagy in female infertility related to aged oocytes and the implication of oxidative stress in autophagy defects in age-related ovarian dysfunction. However, the role of autophagy in embryo implantation in the uterus remains largely unknown. During the menstrual cycle, the autophagic levels change dynamically. The autophagic level in normal ESCs is significantly higher in the secretory phase than in the proliferative phase [14]. However, in patients with endometriosis, ESCs in ectopic endometriosis foci show a constant autophagic level during the menstrual cycle [25]. Therapeutic approaches that inhibit or enhance autophagy have been reported as effective options in experimental endometriosis using rodent models [24,88,89,90]. Thus, whether the autophagic level is higher in normal or endometriotic tissues and whether the therapeutics induce or block autophagy are still being debated [91]. Although endometriosis is closely related to female infertility, in this review we focus on the role of autophagy in embryo implantation.

Among the processes of orchestrated events that are necessary for a successful pregnancy, two of the most critical steps are receptive endometrium and decidualization, which are required for maternal interactions with the developing embryo [71]. High-fat diet-induced obesity and palmitic acid treatment impair the decidualization of ESCs by reducing AMPK and ULK1 expression and decreasing autophagic flux [92]. Deficiency of folate, a major risk factor for birth defects, reduces the autophagy of endometrial cells, thereby inhibiting the apoptosis of decidual cells, restraining endometrial decidualization, and impairing early pregnancy [93].

Several systemic knockout studies have revealed that various ATG-related genes, including *BECN1* (Beclin1), *RB1CC1* (FIP200), and *AMBRA1*, are embryonically lethal with developmental defects [94,95,96,97]. However, the effects of these genes on embryo implantation have not been sufficiently investigated. Recent studies using genetic abrogation have shown that the autophagy of endometrial cells is closely involved in embryo implantation and decidualization [98,99]. Oestreich et al. [98], using a reproductive tract conditional knockout mouse model of *RB1CC1,* revealed that the autophagy protein FIP200 plays a key role in the development of ESCs to decidualized ESCs. They also demonstrated that Atg16L1 is necessary for proper decidualization and blastocyst implantation using mice with a hypomorphic allele of the *Atg16L1* gene (causes a partial loss of function) [99]. In addition, cysteine-rich transmembrane BMP regulator 1 (CRIM1) functions as a regulator of endometrial receptivity at least in part by facilitating Atg7-dependent autophagy in the goat endometrium [100].

Pharmacological autophagy regulators have been examined to determine whether they affect the function of endometrial cells. Rapamycin, an autophagy inducer, reverses the impairment of endometrial decidualization in folate-deficient pregnant mice by disrupting AMPK/mTOR signaling [101]. In addition, Su et al. [16] suggested that autophagy is associated with endometrial decidualization during early pregnancy by revealing impaired uterine decidualization and reduced reproductive rate in female mice treated with the autophagy inhibitors 3-MA and chloroquine. Moreover, zearalenone, a mycotoxin isolated from several *Fusarium* species, blocks autophagic flux by inhibiting the fusion of autophagosomes and lysosomes, inducing the apoptosis of endometrial cells, and ultimately leading to the failure of embryo implantation in young female pigs [102]. Collectively, these reports indicate that endometrial autophagy is essential for embryo implantation, thereby playing a crucial role in endometrial receptivity, decidualization, and subsequent fertility sustenance during early pregnancy (Figure 4).

## 6. Potential Involvement of Autophagic Regulation on the Effect of Natural Products as Embryo Implantation Enhancer

As summarized above, autophagy is increased in the secretory phase of the menstrual cycle and plays a key role in embryo implantation by inducing changes in the uterine endometrium, including endometrial receptivity and decidualization. However, the roles of autophagy-enhancing natural products have not been thoroughly investigated to improve embryo implantation. To date, various natural compounds have been screened and reported as regulators of autophagy [29,30]. Among the various natural compounds that have been identified as autophagy activators, we selected and organized only natural products that have been proven to be safe, are approved by the United States Food and Drug Administration (FDA), and have effects on female fertility (Table 1). Although the direct mechanisms underlying the correlation between autophagy activation and the efficiency of female fertility are still unclear, 20 natural products have been suggested as inhibitors or enhancers of female fertility and have been shown to activate autophagy.

Among these autophagy inducers derived from natural products, 10 compounds, berberine, brefeldin A, curcumin, chrysin, fisetin, α-mangostin, paeoniflorin, rapamycin, γ-tocotrienol, and ursolic acid, have been reported to enhance the female fertility rate by reducing polycystic ovary syndrome (PCOS) and ovarian cell death, protecting the ovary from damage, and improving embryo quality and ovarian life span [103,104,105,106,107,108,109,110,111,112,113,114,115,116,117]. Several of the 10 natural compounds that improve fertility, including berberine, paeoniflorin, ursolic acid, and deferoxamine, have been reported to ameliorate endometriosis [103,118,119,120,121]. Endometriosis is a major cause of female infertility and is associated with reduced oocyte quality and implantation failure [122,123].

Of the 20 natural products that we examined, only nine were found to be directly related to embryo implantation. Four compounds, apigenin, curcumin, genistein, and quercetin, were identified as antagonists for embryo implantation [124,125,126,127,128,129,130]. Four compounds, berberine, emodin, paeoniflorin, and γ-tocotrienol, increased implantation rates by increasing endometrial receptivity or decidualization [103,116,120,131,132]. In contrast, resveratrol has been shown to have dual effects as an agonist and antagonist [133,134]. Kuroda et al. [135] compared these reports and concluded that the timing of drug treatment is important in modulating the decidual response. Resveratrol treatment during the initial decidual phase (i.e., coinciding with the implantation window in vivo) inhibits decidual transformation. However, after the initial phase, resveratrol may promote decidualization by inhibiting decidual senescence. Collectively, the compounds berberine, emodin, paeoniflorin, and γ-tocotrienol might be potential candidates that can increase the rate of embryo implantation, although their effectiveness still requires further investigation.

Several studies have been conducted on the pharmacological or genetic abrogation of autophagy [98,99,100,101], and the role of autophagy in embryo implantation has been largely elucidated. However, the effects of autophagic activators on embryo implantation are not uniform, although the pathways of autophagy activation by these compounds are similar. There are several possible reasons for this heterogeneity. First, the off-target effects of natural products may be different and result in different outcomes. Second, other factors may exist in the “natural products-autophagy activation-embryo implantation” axis. Finally, incomplete autophagic flux may be a source of inconsistency. For example, zearalenone increases Lc3 activation and autophagosome formation, but blocks autophagic flux, thereby leading to implantation failure in gilt [102]. Therefore, further studies should be conducted to develop novel drug candidates to reduce implantation failure by inducing autophagy.

**Table 1 pharmaceuticals-15-00053-t001:** Effects of natural product autophagy regulators on female fertility.

Classification	Name	Chemical Structure	Biological Action	Autophagy-Related Mode of Action	Effect on Female Reproduction	References
Acetohydroxamic acids	Deferoxamine	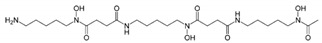	Antibacterial and heavy metal antagonist	mTOR inhibition; elevation of LC3B expression	Protects endometrial stem cells from oxidative damage	[118,119,136]
Alkaloid	Berberine	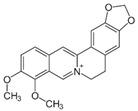	Antioxidant, anticancer, atheroprotective, and immune modulator	Activation of Beclin1; mTOR inhibition	Improves ovulation and endometrial receptivity	[103,137,138,139]
Anthraquinone	Emodin	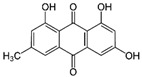	Antioxidant, antidiabetic, and anticancer	Elevation of LC3-II expression	Increases the MET of the endometrial stromal cell (decidualization)	[131,132,140,141]
Flavonoid	Apigenin	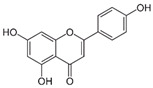	Antioxidant and anticancer	mTOR inhibition	Protects the ovary from ischemic/reperfusion and chemotherapy;antagonizes to progesterone; inhibits embryo implantation	[124,142,143,144]
Chrysin	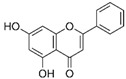	Antioxidant, neuroprotective, and anticancer	Reduction in LC3-II, Beclin1, and ATG7 levels	Protects the ovary from ischemic/reperfusion	[110,145,146]
Fisetin	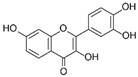	Antioxidant, neuroprotective, and anticancer	mTOR inhibition; AMPK activation	Reduces PCOS	[111,147,148]
Genistein	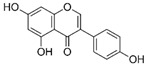	Antioxidant, anti-inflammatory, and anticancer	Inhibition of PI3K-AKT; enhancement of TFEB activity	Induces implantation failure in neonate mice, but not in puberty	[126,127,128,149]
Kaempferol	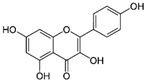	Antioxidant, neuroprotective, and anticancer	AMPK activation	Increases follicle development;activates progesterone signal; relaxes uterine smooth muscle	[150,151,152,153,154,155]
Quercetin	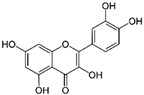	Antioxidant, antiviral, and anticancer	Induction of ATG5 and AMPK activation	Improves follicular development and oocyte quality;inhibits embryo implantation	[129,130,156,157,158]
Wogonin	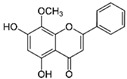	Antioxidant, neuroprotective, anti-inflammation, and anticancer	Induction of ER stress; elevation of LC3-II and Beclin1 levels	Relaxes uterine smooth muscle	[159,160,161]
Lactone	Rapamycin	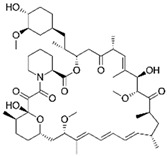	Antibacterial, anticancer, and immunosuppressant	mTOR inhibition	Increases ovarian lifespan	[115,162,163]
Brefeldin A	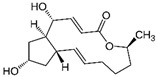	Antiviral and protein transport inhibitor	Enhancement of Bip/AKT activation; reduction in AKT phosphorylation	Increases the survival of female germ cells	[104,105,106,107,164]
Lignan	Magnolol	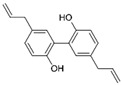	Antioxidant, antidiabetic, and anticancer	mTOR inhibition	Inhibits uterine smooth muscle contraction	[165,166,167,168]
Polyphenol	Curcumin	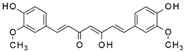	Antioxidant, antidiabetic, antiallergic, and anticancer	Inhibition of mTOR; enhancement of TFEB activity and LC3 levels	Reduces PCOS and POF;inhibits decidualization	[108,109,125,169,170,171]
EGCG, catechin, and epicatechin	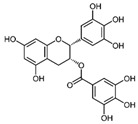 EGCG	Antioxidant, neuroprotective, anti-inflammation and anticancer	AMPK activation	Enhance ovulation; reduce cyst formation in PCOS	[172,173,174,175,176]
Stilbenoid	Resveratrol		Antioxidant, neuroprotective, antidiabetic, and anticancer	AMPK activation	Improves oocyte maturation in aged;increases or decreases decidualization	[133,134,135,177,178,179,180]
Terpenoid	Paeoniflorin	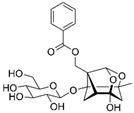	Antioxidant, anti-inflammatory, neuroprotective, and anticancer	LKB1/AMPK activation	Reduces PCOS;enhances endometrial receptivity	[113,114,120,181,182,183]
Ursolic acid	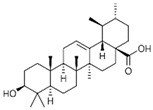	Antioxidant, atheroprotective, antidiabetic, and anticancer	mTOR inhibition; elevation of LC3-II, ATG5, and Beclin1 levels	Attenuates POF (hypothetical);suppresses endometrial stromal cell survival	[117,121,184,185,186]
Tocotrienol	γ-Tocotrienol	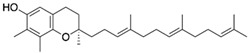	Antioxidant, anti-inflammatory, and anticancer	AMPK activation; elevation of LC3-II, ATG5, and Beclin1 levels	Promotes preimplantation development; improves the quality of embryos	[116,187,188]
Xanthonoid	α-Mangostin	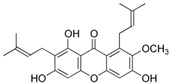	Antioxidant, neuroprotective, and anticancer	AMPK activation; induction of LC3-II	Protects from ovarian cell death	[112,189,190]

Abbreviations: MET, mesenchymal-epithelial transition; PCOS, polycystic ovary syndrome; POF, premature ovarian failure; ATG, autophagy-related gene; mTOR, mammalian target of rapamycin kinase; AMPK, AMP-activated protein kinase; LC3, microtubule-associated protein 1 light chain 3. The chemical structures of compounds were created with ChemDraw (PerkinElmer, Waltham, MA, USA).

## 7. Possible Role of Autophagy on the Effect of Medicinal Herbal Drugs as Embryo Implantation Enhancer

As shown in Table 2, traditional herbal medicines have been used to treat female infertility [191]. Several reports have suggested that traditional herbal formulas and medicinal herbs successfully improve endometrial receptivity and might be an alternative option for improving the outcome of embryo implantation [192,193,194]. Many herbal formulas have been studied for enhancing the embryo implantation rate in animal studies and clinical study [195,196,197,198,199,200,201,202,203,204,205,206,207,208,209,210,211,212,213]. Among these formulas, Erbu Zhuyu decoction and Tokishakuyakusan (TJ-23, Danggui Shaoyao san in Chinese) have been reported to induce autophagy via the expression of Beclin1 and LC3 [208,214,215]. However, as the induction of autophagy in Tokishakuyakusan was not examined in the uterine endometrium, it is not clear whether autophagy activation is directly related to the enhancement of endometrial receptivity.

Medicinal plants have also been studied to improve the rate of embryo implantation. Decusirol isolated from *Angelica gigas* enhanced endometrial receptivity [216] and *Theobroma cacao* and American ginseng increased preimplantation potential without any reproductive toxicity [217,218]. *Theobroma cacao* was reported as an autophagic inducer through the activation of sirtuin-1/AMPK signaling in liver and kidney cells [219,220]. However, decusirol blocked autophagic flux by suppressing the expression of the lysosomal enzyme cathepsin C in gastric cancer cells [221]. Our previous studies showed that several medicinal plants, including *Cnidium officinale*, *Cyperus rotundus*, *Paeonia lactiflora*, *Perilla frutescens* var. *acuta*, and *Rehmannia glutinosa* var. *purpurae*, have a positive effect on improving endometrial receptivity by increasing LIF expression and integrin adhesion molecules [222,223,224,225]. Among these, perillaldehyde from *Perilla frutescens* var. *acuta*, and catalpol from *Rehmannia glutinosa* var. *purpurea* induces autophagy by activating AMPK signaling [226,227,228]. *Cyperus rotundus* is also known to induce autophagy by increasing LC3 and Beclin1 expression [229]. The major active compound responsible for autophagy activation has not yet been elucidated. In particular, the extract from the roots of *P. lactiflora* Pall. improved endometrial receptivity [225]. Paeoniflorin is the main active ingredient of *P. lactiflora* for increasing endometrial receptivity via LIF expression [120]. In addition, paeoniflorin has been reported to induce autophagy and AMPK activation in several types of cells [114,230,231].

These studies have shown that diverse medicinal plants and herbal formulas enhance endometrial receptivity and induce autophagy. The proposed autophagic mechanisms of medicinal formulas and herbal drugs, which are reported as enhancers of embryo implantation, are summarized in Figure 5. However, there is direct evidence that autophagy is directly related to the implantation-promoting effects of these herbal formulas and medicinal herbs. Only Erbu Zhuyu decoction was examined for the expression of autophagic proteins, Beclin1 and LC3B, in the uterine endometrium. In addition, the major active compounds of medicinal compounds that increase autophagy are largely unknown. Diverse natural products used for female infertility have potent antioxidant activity [232], therefore the reduction in oxidative stress could be a possible mechanism underlying their autophagic regulation and improved embryo implantation. The expression of oxidative stress-related genes is involved in idiopathic recurrent miscarriage [233]. In patients suffering from RIF, downregulation of sirtuin-1, a regulator of ROS homeostasis, impeded endometrial decidualization [234]. Resveratrol, an autophagy inducer, restored zearalenone-induced impaired decidualization through induction of the antioxidative gene glutathione peroxide 3 [235]. However, although oxidative stress can increase autophagy initiation, autophagy also contributes to the clearance of irreversibly oxidized molecules [42]. Thus, the relationship between oxidative stress and autophagy remains controversial. To improve our understanding of the mode of action of these herbal medicines and to develop novel therapeutic options to enhance endometrial receptivity, more intensive studies should be performed from the viewpoint of autophagy.

**Table 2 pharmaceuticals-15-00053-t002:** Effect of traditional herbal medicines improving endometrial receptivity on autophagy.

Name	Active Components	Role in Autophagy	References
BaelanChagsangBang	-	-	[236]
Bangdeyun and its component DS147	-	-	[196,197]
Buganshen recipe	-	-	[198]
BuShenAnTai recipe	-	-	[201]
Bushen Tiaoxue Granules and Kunling Wan	-	-	[202]
Dingkun Pill	-	-	[206,207,212]
Erbu Zhuyu decoction	-	Increases the Beclin1 and LC3B	[204,208]
Gushen’antai pills	-	-	[195]
Liuwei Dihuang Granule	-	-	[199]
Shoutaiwai recipe	-	-	[200]
Tokishakuyakusan(Danggui Shaoyao san)	-	Induces autophagy and mitophagy via increasing PINK1 and LC3 but reducing p62	[203,214,215]
Wenshen Yangxue decoction	-	-	[205,211]
Xianziyizhen Recipe	-	-	[237]
Yeosin-san	*Paeonia lactiflora* and *Cyperus rotundus*	-	[213,238]
Yiqixue buganshen recipe	*-*	-	[198]
Zhuyun recipe	-	-	[210]
*Angelica gigas*	Decusirol	Block autophagic flux by suppressing cathepsin C expression	[216,221]
*Cnidium officinale*	-	-	[239]
*Cyperus rotundus*	-	Increases LC3B II/LC3B and Beclin1	[223,229]
*Paeonia lactiflora*	Paeoniflorin	Induces autophagy via inhibition of AKT/mTOR	[114,225,230]
*Panax quiquefolius* (American Ginseng)	Ginsenoside Rb1 and Rg1	Induces autophagy via inhibiting AKT/mTOR	[218,240,241]
*Perilla frutescens* var. *acuta*	Perilaldehyde	Induces autophagy via activating AMPK	[222,228]
*Rehmannia glutinosa* var. *purpurea*	Catalpol	Induces autophagy via activating AMPK	[224,226,227]
*Theobroma cacao*	-	Induces autophagy via activating sirtuin-1/AMPK signaling	[217,219,220]

Abbreviations: mTOR, mammalian target of rapamycin kinase; AMPK, AMP-activated protein kinase; LC3, microtubule-associated protein 1 light chain 3.

Although traditional herbal medicines are generally regarded as safe for use in the clinic, the scientific basis for safety issues is still insufficient. Several herbal medicines and essential oils have been reported to cause harmful adverse outcomes, including fetal resorption, teratogenicity, and embryo-fetotoxicity after maternal exposure [242,243]. Among the medicinal herbs listed in Table 2, Chinese Angelica, a root of the *Angelica* genus, white Paeony root, a root of *P. lactiflora*, and β-elemene, a compound contained in *C. rotundus*, have been reported to increase the rate of adverse effects. Several herbal drugs, such as *Lippa citriodora* leaves and large head Atractylodes roots, have shown teratogenic adverse effects [244,245]. In contrast, another study reported that white Paeony root does not induce any harmful adverse effects up to the highest dose tested of 32 g/kg/day [246]. In addition, safety assessment of products containing Angelica extract demonstrated no adverse effects when used during pregnancy [247,248]. Moreover, the whole extract of Korean ginseng and American ginseng does not increase harmful adverse effects in pregnant mice [218,249], although ginsenoside Rb1 retarded early pre- and post-implantation development of mouse embryos by inducing ROS-mediated apoptosis [250]. To precisely evaluate safety issues, further good laboratory practice (GLP)-level reproductive toxicity studies should be conducted. Currently, caution should be exercised in the clinical use of medicinal herbal drugs during pregnancy.

## 8. Conclusions and Perspective

Successful pregnancy requires a sequence of orchestrated events, including embryo implantation and decidualization [18,71]. These steps are critical for maintaining early pregnancy because they mediate the interaction between the maternal uterus and the developing embryo. If the embryo implantation and decidualization processes are unsuccessful or improperly regulated, it may result in the loss of early pregnancy, RIF, or pre-eclampsia [18,70]. Recent studies have demonstrated that autophagy is elevated in the secretory phase of the menstrual cycle and is positively related to the formation of the receptive endometrium [15,16]. Additionally, the decidualization process is mediated by a highly regulated autophagy process [101]. Thus, autophagy may be a therapeutic target for improving embryo implantation.

Diverse natural products have been reported to be autophagy inducers, and many researchers are attempting to find novel therapeutic agents for the treatment of autophagy-defective diseases, including cancer, neurodegenerative diseases, and aging [29,30,251]. Thus, natural products can be applied to improve the embryo implantation rate in an autophagy-dependent manner. However, to date, only a few studies have focused on the topic of autophagy as a major mechanism underlying the improvement of embryo implantation, and thereby the treatment of female infertility by natural products. In addition, the present autophagy inducers showed limitations due to their low specificity, irregular distribution, and rapid clearance [59]. To encourage the development of novel drug candidates for treating autophagy-related female infertility, the following two strategies might be helpful. Formerly, natural product-based autophagy inducers could be structurally modified to enhance efficacy, specificity, bioavailability, and safety [252,253]. Nano-delivery of natural products might be an option to improve aqueous solubility, bioavailability, and distribution to specific tissues [59,254]. Several studies have shown that natural product-loaded nanoparticles, including the autophagy inducers resveratrol, quercetin, and curcumin are promising for the treatment of various diseases such as cancer, inflammation, and arthritis [255]. Toxicity concerns should be addressed before their clinical use in female infertility patients.

There are several reasons for the limited scope of this review. First, the mechanism of embryo implantation is still largely unknown, particularly its relationship with autophagy. Second, there are not many pro-autophagic drug candidates that are safe for use in early pregnancy. Third, in vivo methods for monitoring autophagic flux have not been fully developed. Thus, these limitations should be addressed in future studies to elucidate the precise role of autophagy in embryo implantation and to identify potential candidates for treating female infertility from the perspective of endometrial receptivity and decidualization in early pregnancy.

## Figures and Tables

**Figure 1 pharmaceuticals-15-00053-f001:**
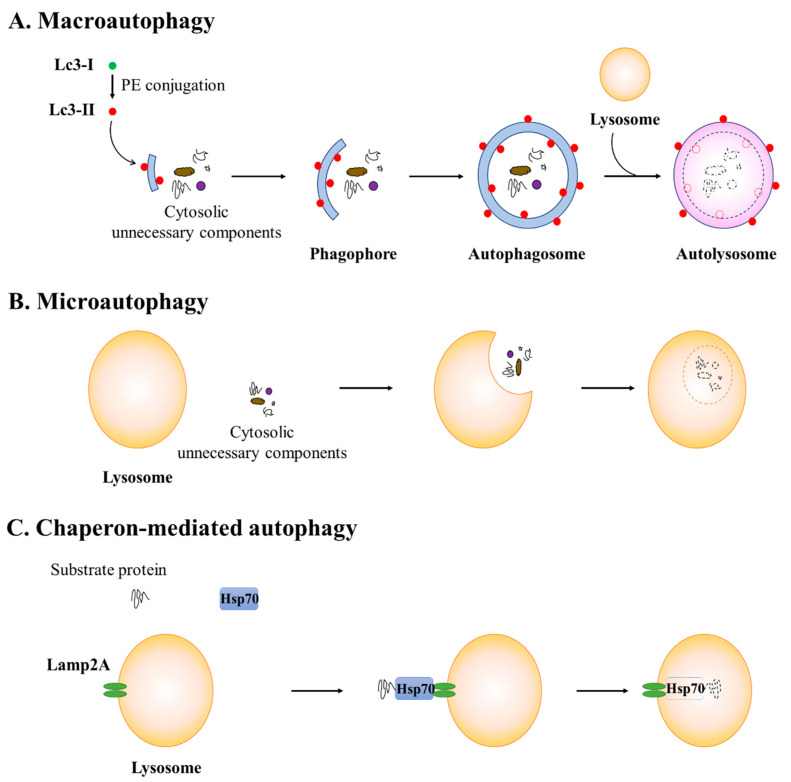
Three types of autophagy. There are three types of autophagy, depending on the cargo delivery system to the lysosome. (**A**) In macroautophagy, cytosolic components are sequestered within autophagosomes, which subsequently fuse with lysosomes. (**B**) By contrast, in microautophagy, lysosomes directly sequester the cytosolic components. (**C**) In chaperone-mediated autophagy, heat shock protein 70 chaperon (Hsp70) recognizes substrate proteins and delivers them to lysosomal-associated membrane protein type 2A (Lamp2A) in the lysosome membrane. The substrate proteins are translocated to the lysosomal lumen for degradation by lysosomal enzymes. Abbreviations: Atg, autophagy-related gene; Lc3, microtubule-associated protein 1 light chain 3; PE, phosphatidylethanolamine.

**Figure 2 pharmaceuticals-15-00053-f002:**
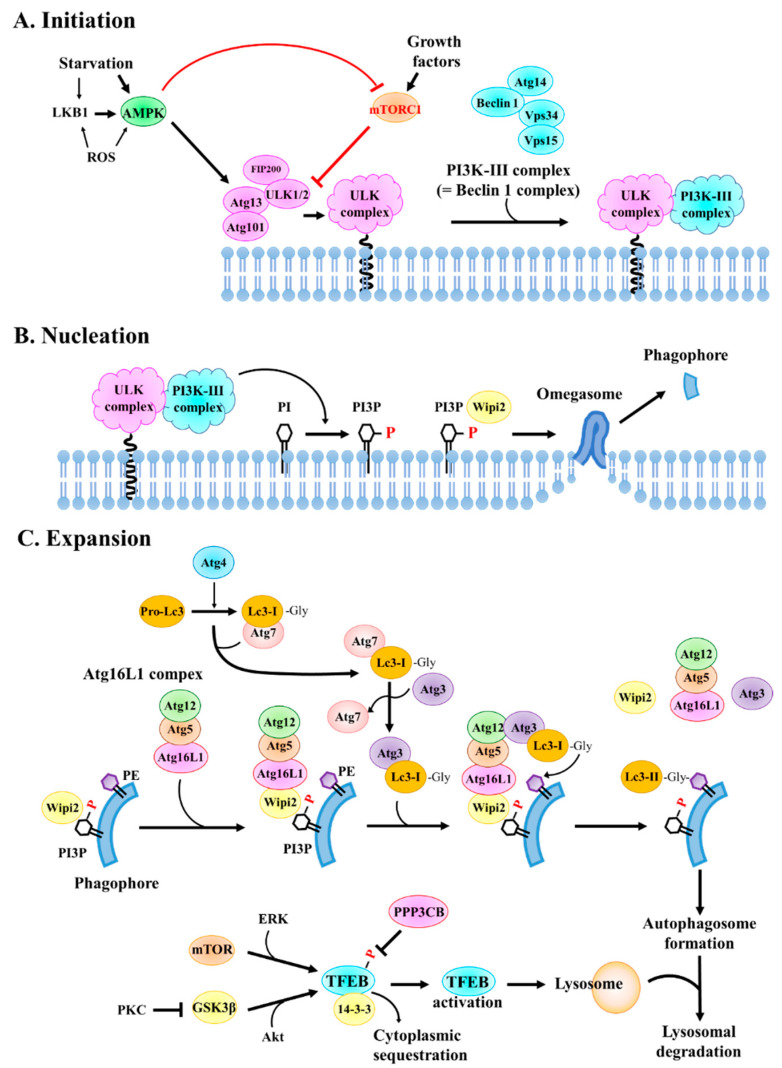
Autophagosome formation. Autophagosome formation can be divided into three stages. (**A**) During initiation, the UNC51-like kinase (ULK) complex is dissociated from mTORC1 and binds to the autophagosome initiation site. The ULK complex recruits and phosphorylates the class III phosphoinositide 3-kinase (PI3K-III) complex. (**B**) During nucleation, the PI3K-III complex generates phosphatidylinositol 3,4,5-triphosphate (PI3P) on the membrane and recruits autophagy-specific PI3P effectors, such as WD-repeat domain phosphoinositide-interacting 2 (Wipi2). The interaction of PI3P with Wipi2 contributes to the phagophore formation. (**C**) During expansion (elongation), the Atg12-Atg5-Atg16L1 complex (also known as the Atg16L1 complex) is recruited to the membrane and lipidates microtubule-associated proteins 1 light chain 3 (Lc3). The pro-form of Lc3 is cleaved at the carboxyl-terminal (C-terminal) by Atg4 and becomes cytosolic Lc3-I, thereby exposing the C-terminal glycine residue. Lc3-I is subsequently transferred to the autophagosome by Atg3 and conjugated with phosphatidylethanolamine (PE) at the C-terminal glycine residue by the Atg16L1 complex, resulting in the formation of Lc3-II. During the autophagy process, Lc3-II bound to the autophagosomal inner membrane is degraded by lysosomal enzymes. The expression of genes related to lysosomal biogenesis and autophagolysosome formation is controlled by a master transcriptional regulator, TFEB. Abbreviations: AMPK, AMP-activated protein kinase; Atg, autophagy-related gene; ERK, extracellular signal-regulated kinase; LKB1, liver kinase B1; GSK3β, glycogen synthase kinase-3β; mTOR, mechanistic target of rapamycin kinase; PKC, protein kinase C; PPP3CB, protein phosphatase 3 catalytic subunit beta; TFEB, transcription factor EB; Vps, vacuolar protein sorting-associated protein, Lc3, microtubule-associated proteins 1 light chain 3.

**Figure 3 pharmaceuticals-15-00053-f003:**
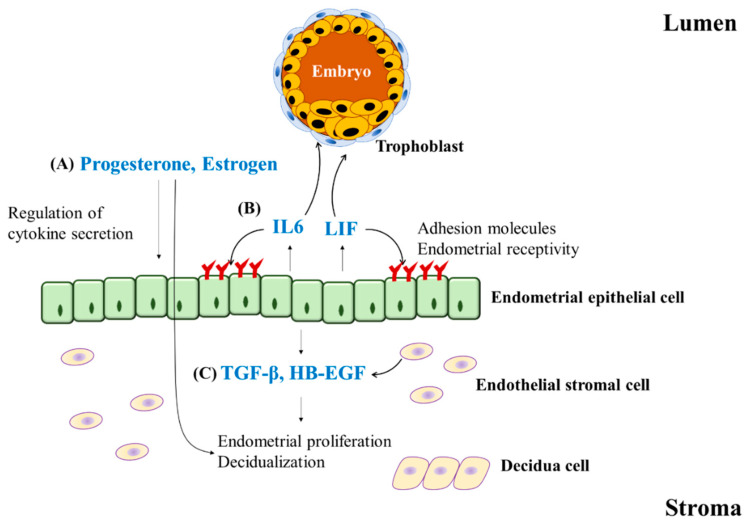
Regulatory factors in embryo implantation. Embryo implantation is regulated by diverse factors. (**A**) The ovarian steroid hormones progesterone and estrogen facilitate the appropriate morphology, function, and development of the endometrium during the implantation period. (**B**) The cytokines leukemia inhibitory factor (LIF) and interleukin 6 (IL6) are involved in the regulation of endometrial receptivity via expressing adhesion molecules, which play a crucial role in the attachment of the trophoblast to the uterine epithelium. (**C**) The growth factors transforming growth factor-β (TGF-β) and heparin binding-epidermal growth factor (HB-EGF) are expressed in endometrial stromal and epithelial cells to regulate endometrial cell proliferation and decidual transformation.

**Figure 4 pharmaceuticals-15-00053-f004:**
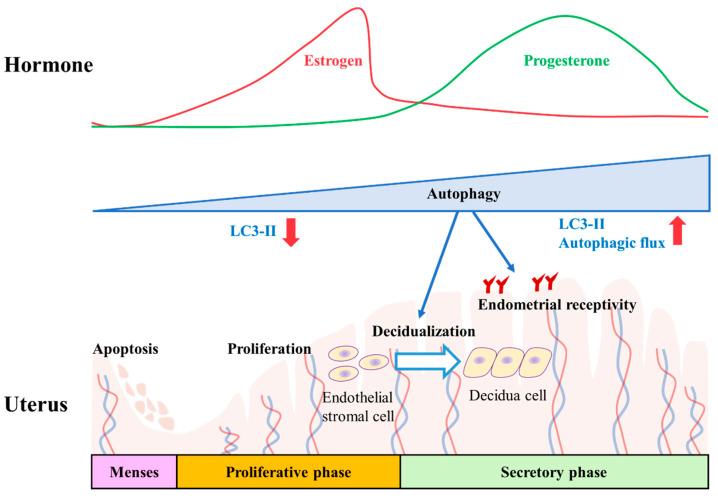
Role of autophagy in embryo implantation. Cyclic changes in ovarian steroid hormones, including estrogen and progesterone, regulate the growth, differentiation, and apoptosis of endometrial cells in the different phases of the uterine endometrium. The levels of Lc3-II and flux of autophagy are increased in the secretory phase, correlating with the level of progesterone. Defects of autophagy directly affect the receptivity of endometrial epithelium and decidualization of endometrial stromal cells. Abbreviation: LC3, microtubule-associated proteins 1 light chain 3.

**Figure 5 pharmaceuticals-15-00053-f005:**
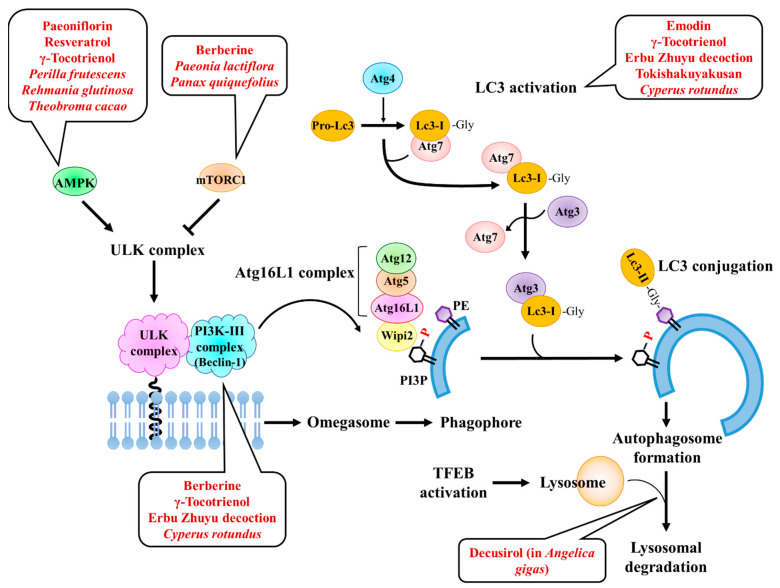
The proposed mechanisms of natural products that enhance embryo implantation. Paeoniflorin, resveratrol, γ-Tocotrienol, *Perilla frutescens*, *Rehmannia glutinosa*, and *Theobroma cacao* activate AMPK and thereby induce autophagy. Berberine, *Paeonia lactiflora*, and *Panax quinquefolius* inactivates mTORC1. Berberine, γ-Tocotrienol, Erbu Zhuyu decoction, and *Cyperus rotundus* reduced Beclin1. Emodin, γ-Tocotrienol, Erbu Zhuyu decoction, Tokishakuyakusan, and *Cyperus rotundus* increase LC3 expression and/or its activation. Decusirol isolated from *Angelica gigas* interferes with autophagic flux. Abbreviations: AMPK, AMP-activated protein kinase; Atg, autophagy-related gene; mTOR, mammalian target of rapamycin kinase; Lc3, microtubule-associated proteins 1 light chain 3.

## Data Availability

The data will be made available upon reasonable request.

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
