# Peer review of "Autophagy as a Therapeutic Target of Natural Products Enhancing Embryo Implantation"

_pharmaceuticals, 2021, doi:10.3390/ph15010053_

Round 1

Reviewer 1 Report

The manuscript entitled: "Autophagy as a Therapeutic Target of Natural Products Enhancing Embryo Implantation" mainly focuses on the role of autophagy in infertility and embryo implantation. 

The manuscript is well written and the mechanisms are well described. However, it is lacking in the description of mechanisms via which natural products employ autophagy as the therapeutic target for infertility and embryo implantation. The authors should better describe the mechanism of action of these natural products and propose where future research should aim, in order to validate the efficiency -and perhpas the safety- of these products.

Author Response

I attached the answer to the reviewer's comment as a word file.

Reviewer 2 Report

Park et al reviewed literatures on the relationship between natural products, autophagy and embryo implantation. Although the information from this review might help some readers, there are two major issues to be addressed.

  1. Overall, the current manuscript is not well-written. There are quite a few grammatical errors and many paragraphs are not well organized.
  2. This review is less likely to provide new information. For instance, the sections 2-3 covered very general information without providing new and updated findings. In addition, the section 5, the most important part in this manuscript, provide very limited information.

Author Response

Park et al reviewed literatures on the relationship between natural products, autophagy and embryo implantation. Although the information from this review might help some readers, there are two major issues to be addressed.

1. Overall, the current manuscript is not well-written. There are quite a few grammatical errors and many paragraphs are not well organized.

Answer: Thank you for the constructive criticism. Although the manuscript was edited by a native English speaker, we have had the manuscript reviewed again for language editing before submitting the revised version.

2. This review is less likely to provide new information. For instance, the sections 2-3 covered very general information without providing new and updated findings. In addition, the section 5, the most important part in this manuscript, provide very limited information.

Answer: We agree with the reviewer’s criticism of the volume balance of Sections 2-3 and Section 5. However, to address this special issue, the focus of this manuscript is to discuss the possible usefulness of natural products in embryo implantation by activating autophagy. In addition, there are very few research papers concerning the topic of autophagy and embryo implantation. We have attempted to cite all previous research papers in this manuscript and modified to reflect reviewer’s comment as possible. Your concern might be resolved by accumulating extensive studies in the future. I hope that this manuscript could serve to expand the research topic, “autophagy and embryo implantation”

As a corresponding author, I very much appreciate your kind and constructive comments. The manuscript has been improved by your suggestions.

Reviewer 3 Report

The authors describe the relationship between autophagy and embryo implantation and explore possible therapeutic options based on natural compounds to modulate autophagy and therefore treat female infertility.

The topic is highly innovative, very little is know about the role of autophagy in this process and this review could elicit the community to investigate further this relevant topic.

The manuscript could be published after considering the following points here listed: 

- In table 1 the authors report that genistein and curcumin have effect on TFBE transcription factor but its role is not detailed in the text.  Hence, in section 3: "Regulation of Autophagosome Formation" the authors should emphasize the role of TFBE on the regulation of autophago - lyososomal networks as detailled in the following bibliography doi: 10.3390/cells10102752. This should be also included in the figure 2.

- The part of discussion/conclusion is very short. In this sense, since autophagy modulation by nanoparticles is a novel hot topic with potential applications in medicine, I recommend mention in the conclusions the possibility to employ nanocarriers delivering natural products to modulate autophagy to treat infertility. There is plenty of literature about this topic that may be cited doi: 10.1016/j.nano.2020.102270, doi: 10.1007/s10311-020-01061-2

- Also, the authors should discuss about the connexion between oxidative stress and autophagy (two process strictly interconnected) and how this interplay affects female fertility... doi: 10.1007/s10311-020-01061-2

- Other relevant literature related to the topic should be mentioned and discussed 10.1007/s10311-020-01061-2

- I recommend to insert a figure summarizing the effect of natural compounds on molecular players involved in autophagy. This would semplify the reader.

Author Response

I attached the answer to the reviewer's comment as a file.

Round 2

Reviewer 3 Report

The authors replied to all my concerns and provide a revised version of the manuscript, that can be accepted now

Author Response

The authors replied to all my concerns and provide a revised version of the manuscript, that can be accepted now.

Answer: Thanks for the positive evaluation. We appreciate for reviewer’s constructive comments.